# An inertial sensor-based comprehensive analysis of manual wheelchair user mobility during daily life in people with SCI

Kathylee Pinnock Branford[1], Meegan G. Van Straaten[2], Omid Jahanian[3], Melissa M. B. Morrow[4], Stephen M. Cain[1]*

1 Department of Chemical and Biomedical Engineering, West Virginia University, Morgantown, West Virginia, United States of America, 2 Division of Health Care Delivery Research, Robert D. and Patricia E. Kern Center for the Science of Health Care Delivery, Mayo Clinic, Rochester, Minnesota, United States of America, 3 Department of Physical Medicine and Rehabilitation, Rehabilitation Medicine Research Center, Mayo Clinic, Rochester, Minnesota, United States of America, 4 Department of Physical Therapy and Rehabilitation Sciences, Sealy Center on Aging, The University of Texas Medical Branch, Galveston, Texas, United States of America

* stephen.cain@mail.wvu.edu

## Abstract

This study employed three inertial measurement units to quantify the mobility characteristics of 12 manual wheelchair users with spinal cord injuries (SCI) over 7 consecutive days, revealing nuanced patterns of daily movement. Mobility metrics were calculated for measures of distance traveled, movement duration, and speed. A mobility profile was created to understand patterns of movement behaviors. Participants moved 65.54±21.81 min daily, traveled 1488.15±700.09 meters at an average speed of 0.43±0.16 m/s, and executed approximately 910 turns and 428 starts/stops per day. Mobility predominantly occurred in short bouts (<215 seconds), accounting for 94% of the mobility bouts. Mean mobility characteristics remained consistent across participants despite individual variability in high-resolution metrics, including starts/stops, turns and navigated slopes exceeding the ADA recommended ratio. This methodology provides insights into real-world manual wheelchair mobility and future research could inform rehabilitation strategies and assistive technology development. These methods underscore the critical importance of personalized, high-resolution mobility assessments in understanding and optimizing manual wheelchair users' functional independence and quality of life.

## Introduction

Manual wheelchair (MWC) use is a critical aspect of mobility for many individuals with spinal cord injuries (SCIs) and other disabilities. It is estimated that about 40% of the 1.5 million MWC users in the U.S. are individuals with SCIs [1]. Each year, approximately 18,000 new cases of traumatic SCIs occur in the United States, contributing to

**Data availability statement:** All relevant data are within the manuscript and its Supporting Information files.

**Funding:** This publication was made possible by funding from the Eunice Kennedy Shriver National Institute of Child Health and Human Development (NIH; grant no. R01HD84423, PI: Morrow). NIH had no role in study design, data collection and analysis, decision to publish, or preparation of the manuscript.

**Competing interests:** The authors have declared that no competing interests exist.

a growing population of MWC users [2]. Limited mobility can negatively affect overall health and has been linked to various health conditions, including diabetes and obesity [3–6]. Consequently, understanding the factors that impact mobility and accessibility during daily life is crucial.

Despite the well-documented benefits of an active lifestyle, research indicates that physical inactivity is more prevalent among individuals with disabilities compared to those without disabilities [7]. Reduced mobility can further worsen secondary health complications, such as pressure injuries, obesity, cardiovascular disease, and depression. MWC users face unique challenges to maintain sufficient mobility and physical activity levels due to inadequate accessibility, lack of specialized exercise equipment, and upper extremity pain or fatigue [8]. Several studies have identified environmental barriers that limit the mobility and participation of MWC users, including obstacles related to the environment, such as narrow doorways, steep ramps, and uneven sidewalks. Inaccessible public transportation, lack of accessible housing, and attitudinal barriers can also restrict community participation [9]. Addressing these challenges requires innovative approaches to accurately assess and monitor the mobility of MWC users in their daily environments.

Gaining insights into the daily experiences and challenges faced by wheelchair users requires an understanding of their mobility in real-world environments. Advancements in wearable sensor technologies have enabled researchers to objectively measure these patterns in free-living conditions, collecting data on distance traveled, speed, movement duration, and turns [10–17].

Previous research has established baseline metrics for wheelchair mobility across different environments. In residential settings, wheelchair users travel 1,900−2,457 meters daily at speeds of 0.63–0.79 m/s, accumulating 47.9±21.4 minutes of movement [15]. One study [11] combined objective mobility data with survey-based assessment using the Craig Handicap Assessment Recording Technique to evaluate how wheelchair mobility relates to demographics, wheelchair type, and community participation. In contrast, long-term care facility residents show more limited mobility, traveling only 532±406 meters per day at similar speeds (0.76±0.18 m/s) with 10.8±6.9 minutes of movement per hour [18].

Research has also introduced the concept of "bouts of mobility" to analyze movement patterns [12]. The median bout lasts 21 seconds, covers 8.6 meters at 0.43 m/s, with 63% of bouts shorter than 30 seconds and 85% lasting 60 seconds or less. Recent work incorporating turn analysis found that wheelchair users make approximately 900 turns per day, with mean daily distances of 3.10 km [14]. Additional studies have examined physical activity levels [19–21] and movement in wheelchair sports [19–22]. While existing research has established fundamental wheelchair mobility metrics—including bout duration, distance, turning frequency, and velocity—through sensor-based measurements, a critical gap remains in understanding how these movement characteristics vary across different environmental contexts and daily activities. Environmental contexts include diverse settings from indoor spaces (homes, offices, stores) to outdoor areas (sidewalks, parks, campuses), each with distinct challenges like different surfaces, obstacles, and crowds. Context-specific

motion measurements could reveal how wheelchair users adapt their movement strategies across these varied environments, offering deeper insights than general mobility metrics. Currently, this knowledge gap limits our ability to develop tailored mobility assistance technologies and environmental interventions that accurately reflect real-world wheelchair usage patterns across different contexts.

To bridge this gap, our study built upon previous research by developing a comprehensive analysis of MWC mobility by employing data measured by IMUs to assess and quantify detailed characteristics of MWC use during daily life. Such an approach can be used to understand modifiable factors that impact MWC mobility and their relationship to daily arm use for mobility. Our analysis includes traditional mobility metrics including distance traveled, movement duration, and mean speed. Additionally, we incorporated a more detailed analysis of MWC maneuvering, including the number of turns, starts, stops, into a novel approach that we call a mobility profile. A mobility profile provides a weekly summary that quantifies maneuvering behaviors, categorizes bout durations using data-driven thresholds (short < 215s, medium 215s-700s, long > 700s), and assess environmental demands to provide comprehensive insight into movement patterns through cluster analysis. For each bout of mobility, we calculated bout continuity (ratio of time spent moving to total bout time), which has been identified as important for understanding indoor versus outdoor pedestrian mobility [23], and identified if the MWC user navigated any slopes exceeding ADA guidelines. Our comprehensive analysis provides a more nuanced understanding of wheelchair mobility in everyday environments. Understanding mobility patterns can inform the development of tailored interventions and assistive technologies to enhance the mobility and participation of individuals who rely on MWCs.

## Materials and methods

### Participants

This study was approved by the Mayo Clinic Institutional Review Board (15–004974). Participants were recruited as part of a larger longitudinal investigation titled "Natural History of Shoulder Pathology in Wheelchair Users" (NCT02600910). The recruitment period started July 9, 2015, and ended January 27, 2020 (09/07/2015–27/01/2020). Eligible individuals were between 18 and 70 years of age, had sustained a spinal cord injury (SCI), and utilized a MWC as their primary means of mobility. Written informed consent was obtained from all participants before enrollment. Real-world data was collected from a cohort of 12 MWC users with SCI (Table 1) over seven consecutive days, as part of an observational, longitudinal study focused on wheelchair mobility and arm use in daily life. Participants had a mean age of $47 \pm 13.4$ years and a mean time since SCI of $21 \pm 13.7$ years. SCI levels were classified into three groups: high (C6-C8), mid (T1-T8), and low (T9-L1).

Table 1. Subject demographics.

| Participant | Age Category | Sex | Time since Injury Years | SCI level | Employment status |
|---|---|---|---|---|---|
| P1 | 30-39 | Male | 9 | Mid | Part-time (on-site) |
| P2 | 40-49 | Female | 8 | Mid | Part-time (remote) |
| P3 | 30-39 | Male | 8 | Mid | Part-time (on-site) |
| P4 | 50-59 | Male | 27 | Low | Retired |
| P5 | 50-59 | Male | 30 | High | Part-time (remote) |
| P6 | 50-59 | Male | 28 | Mid | Self-employed (hybrid) |
| P7 | 50-59 | Female | 39 | Low | Self-employed (hybrid) |
| P8 | 30-39 | Male | 12 | High | Full-time (on-site) |
| P9 | 30-39 | Male | 10 | Mid | Full-time (on-site) |
| P10 | 40-49 | Male | 18 | High | Full-time (on-site) |
| P11 | 60-69 | Male | 49 | Mid | Retired |
| P12 | 50-59 | Female | 10 | Low | Full-time (on-site) |

## Free-living data collection protocol

We utilized three IMUs (Axivity-AX6; range ±16 g and ±2000 deg/s, sampling frequency 100 Hz, weight 11g) that were placed on the MWC (frame and wheels) (Fig 1). The IMUs used in this study allowed seven days of continuous data recording without recharging when operating in full IMU mode (recoding both linear acceleration and angular velocity) recording at 100 Hz.

This study aimed to capture data representative of the participants' typical routines, requiring 7 days of data to ensure comprehensive monitoring without disrupting normal behavior patterns [24]. Sensors were secured on the wheelchair for the entirety of the week-long data collection, as no charging was required. Participants were asked to begin each day with a wheelchair pivot movement (Fig 2), creating synchronization data (angular velocity about a vertical axis) for use in

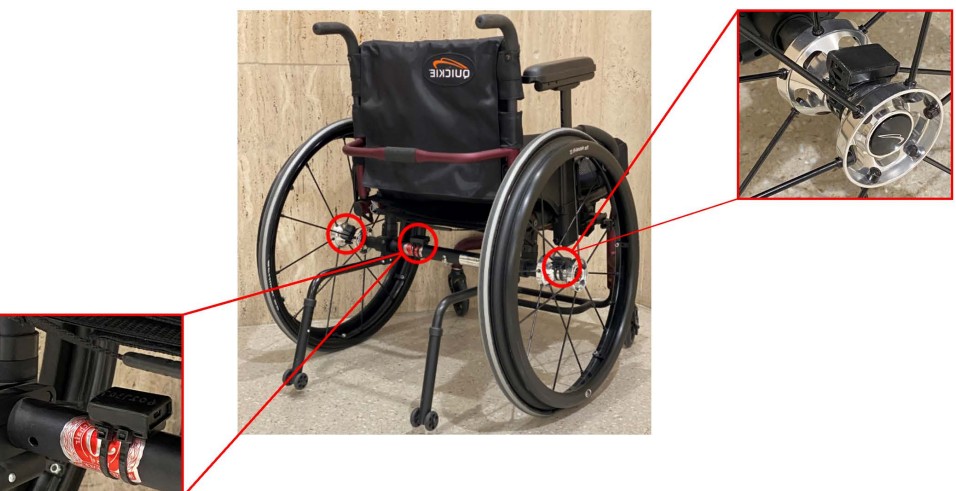

**Fig 1. Inertial measurement unit (IMU) placement on MWC.** The image illustrates the placement of the three MWC-mounted IMUs. The IMUs were secured with custom 3D-printed holders. Red circles indicate the mounting locations at both wheel hubs and the wheelchair frame.

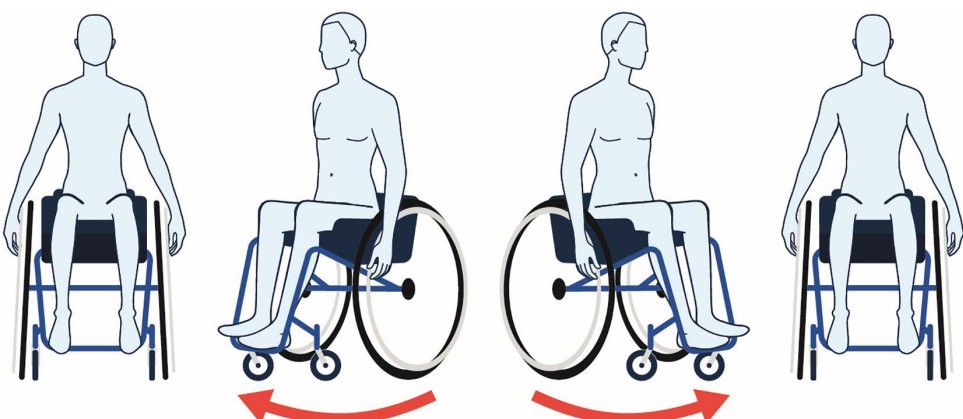

**Fig 2. Wheelchair Pivot Maneuver for IMU Synchronization.** Schematic illustration of a wheelchair pivot maneuver. The sequence shows four positions during the maneuver: starting position (left), clockwise rotation (middle-left), counterclockwise rotation (middle-right), and return to starting position (right). Red arrows indicate the direction of rotation. This standardized movement pattern creates distinct angular velocity measurements across all IMUs.

post-processing to correct for time drift between sensors [22,25,26]. Additionally, participants were asked to continue their daily activities as usual and not to change their routines. Upon completion of the data collection period, a research team member retrieved each set of sensors during an in-person visit. After data download, data were visually inspected using the manufacturer's software to confirm signal quality and completeness. Days beyond the designated 7-day period were excluded from analysis.

## Data processing and variable extraction

Once the sensors were returned to the research team, data were downloaded from the IMUs onto a computer and processed (Fig 3). Sensor data were segmented by days. Next, the data were synchronized using the chair pivoting timing throughout the week by using cross-correlation to calculate the time shift between sensors using the measured angular velocities about a vertical axis (inertial reference frame) from each sensor (for more information refer to S1 Appendix). For each day, sensor-to-wheelchair alignment and orientation were calculated for each sensor.

**Orientation calculations.** Orientation quaternions were calculated from the raw accelerometer and gyroscope data using an open-source algorithm [27,28] designed for efficient inertial motion tracking. The algorithm employs a quaternion-based gradient descent method that estimates orientation by minimizing the error between measured and expected acceleration and magnetic field directions. This method enables accurate orientation estimation by correcting for gyroscope drift using accelerometer and magnetometer data. For this study, magnetometer data were not used. The algorithm relied on accelerometer and gyroscope signals to compute the quaternion orientation for each sensor across the recording period.

These quaternions were subsequently converted to direction cosine matrices to facilitate coordinate frame transformations [29]. The direction cosine matrix that defines the transformation of measurements made in the sensor-fixed frame to those in the inertial frame was defined as $R_{inertial|IMU}$.

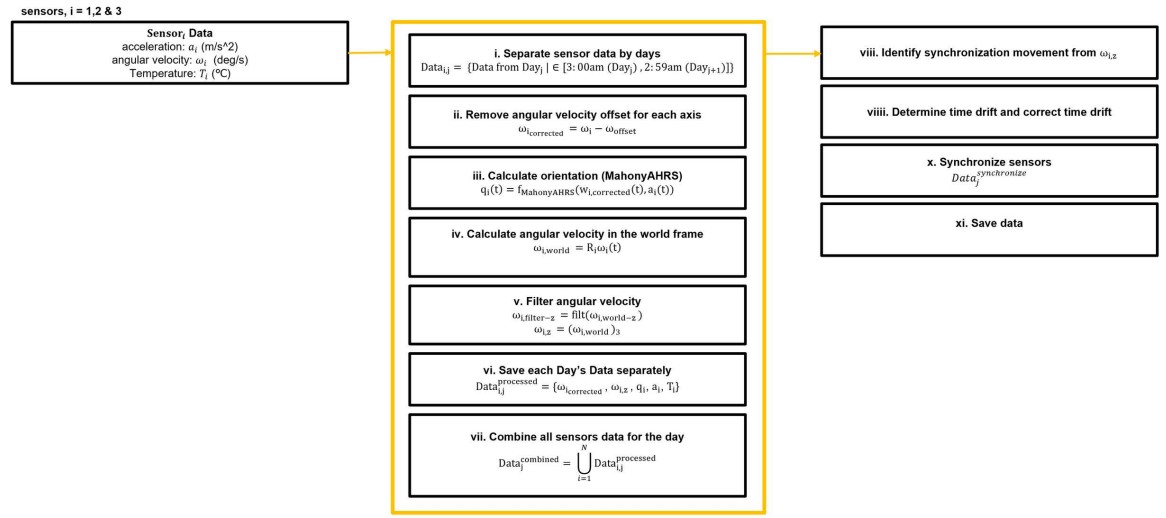

**Fig 3. Data processing for multi-sensor synchronization.** i. Raw sensor data from 3 sensors is separated into daily sections, from 3:00 am of the current day to 2:59 am of the next day, **ii.** Angular velocity measurements are corrected by removing any systematic offset for each axis, **iii.** Orientation quaternions were computed using an open-source gradient descent algorithm [26] that fuses accelerometer and gyroscope data to estimate sensor orientation in a computationally efficient manner. **iv.** Angular velocity is transformed into the inertial frame using the calculated orientation, **v.** angular velocity is filtered **vi.** Processed data for each day is saved separately, containing corrected angular velocity, acceleration, world-frame z-axis angular velocity, orientation quaternions, and temperature, **vii.** All processed sensor data for the day is combined into a single dataset, **viii.** Synchronization movements are identified using the world-frame angular velocity, **viiii.** Temporal drift between sensors is determined using cross-correlation and corrected, **x.** Data is synchronized across sensors, **xi.** Save data.

**Sensor-to-wheel alignment.** Each wheel's axis of rotation was defined with a principal component analysis (PCA) of the measured wheel angular velocity. The first principal component, which captures the majority of variation in measured angular velocity, is used to define the axis of rotation and is aligned with the wheel axle direction. Wheel rotation velocity is calculated by taking the dot product of the measured angular velocity and the wheel axis of rotation ($\omega_{wheel,i}$ where i = R or L for right or left wheel).

The linear velocity of each wheel was calculated by multiplying the wheel rotation velocity (in rad/s) with the wheel radius, which was derived from the measured wheel circumference. Wheelchair speed ($v_{WC}$) was the average of the calculated linear velocities of both wheels. Wheelchair linear acceleration ($a_{WC}$) was obtained by numerically differentiating wheelchair speed. No filtering was applied prior to differentiation.

**Sensor-to-wheelchair alignment.** For the IMU (Fig 4A) secured to the wheelchair frame, we defined two reference frames: (1) a wheelchair-fixed frame (Fig 4B), which is fixed to the wheelchair (and therefore the IMU) with axes aligned with the wheelchair and (2) an inertial frame (Fig 4C), which is fixed relative to gravity and an arbitrary heading direction.

First, we defined the wheelchair-fixed frame ($\hat{X}_{WC}$, $\hat{Y}_{WC}$, $\hat{Z}_{WC}$). The average acceleration due to gravity measured during no movement ($\vec{a}_{no\ movement}$) was used to define a wheelchair-fixed z-axis ($\hat{Z}_{WC}$) assuming that the axis is aligned with gravity [30] and that the wheelchair is positioned on level ground:

$$\hat{Z}_{WC} = \frac{\vec{a}_{no\ movement}}{\sqrt{\vec{a}_{no\ movement} \cdot \vec{a}_{no\ movement}}} \tag{1}$$

We defined the forward direction axis ($\hat{X}_{WC}$) by assuming that one sensor axis is already aligned well with a wheelchair axis through proper sensor mounting (Fig 1). Given this mechanical alignment, the forward direction axis is calculated as:

$$\hat{X}_{WC} = \frac{[1\ 0\ 0] \times \hat{Z}_{WC}}{norm\left([1\ 0\ 0] \times \hat{Z}_{WC}\right)} \tag{2}$$

And finally, we ensured that the $\hat{Y}_{WC}$ axis is orthogonal to the $\hat{Z}_{Chair}$ and wheelchair axes:

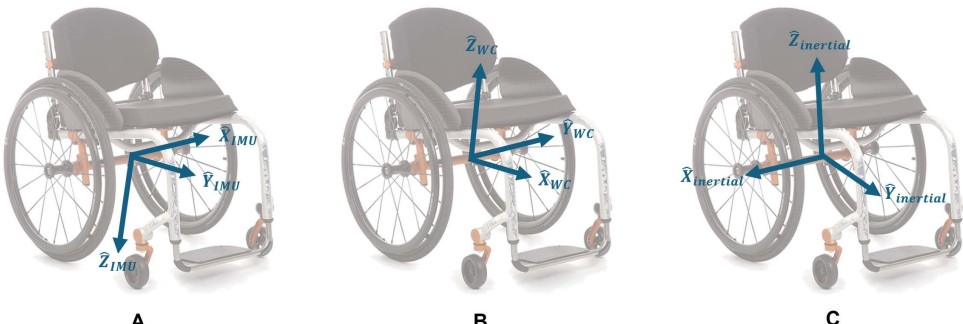

| A | B | C |
|---|---|---|

**Fig 4. Reference frame definitions for IMU-wheelchair alignment.** Illustration of three coordinate reference frames used for IMU data analysis on a manual wheelchair. **(A)** A sensor-fixed frame, which is fixed to and rotates with the IMU. **(B)** A wheelchair-fixed frame is fixed to and rotates with the wheelchair with axes aligned with axes consistent with vehicle dynamics. **(C)** An inertial frame, which is fixed relative to gravity and an arbitrary heading direction and does not rotate with the sensor or wheelchair.

$$\hat{Y}_{WC} = \frac{\hat{Z}_{grav} \times \hat{X}_{WC}}{\text{norm}(\hat{Z}_{grav} \times \hat{X}_{WC})}$$

(3)

The resulting unit vectors ($\hat{X}_{WC}$, $\hat{Y}_{WC}$, $\hat{Z}_{WC}$) define a wheelchair-fixed frame aligned to the wheelchair frame axes. The direction cosine matrix that defines the transformation of measurements made in the sensor-fixed to the wheelchair-fixed frame is defined by Equation (4), where each row of the matrix contains the components of the wheelchair-fixed axes [27].

$$R_{WC \mid IMU} = \begin{bmatrix} \hat{X}_{WC} \\ \hat{Y}_{WC} \\ \hat{Z}_{WC} \end{bmatrix}$$

(4)

To improve estimates of the wheelchair frame orientation, we subtracted the calculated acceleration of the wheelchair ($a_{WC}$) obtained from the wheel's angular velocity from acceleration resolved in the wheelchair-fixed frame. This new acceleration was used to recalculate orientation [27,28].

We utilized wheelchair, sensor, and inertial frame measurements to quantify mobility (Table 2). We calculated wheelchair speed, angular velocity of the MWC about vertical, wheelchair movement trajectories, wheelchair displacement, and tilt of the wheelchair. We used these calculated wheelchair kinematics to identify turns, starts and stops, mobility bouts, slopes, and slopes that exceed the ADA-compliant ratio.

## Mobility metrics

- **Data analysis period duration:** The duration of the data analysis period was measured in hours, spanning from the first detected movement of the MWC to its last movement within the defined 24-hour timeframe (3 AM to 2:59 AM the following day). This choice was made to account for individuals who remain active past midnight, such as those with

**Table 2. Mobility metrics definitions and sensors used.**

| Metric | Definition | Sensor |
|---|---|---|
| Number of turns | Total number of turning maneuvers performed by a manual wheelchair (MWC) user during the data collection period. | WC frame |
| Number of starts and stops | Total number of movement initiations and terminations performed by a MWC user during the data collection period. | Wheels |
| Duration* | Total time of a mobility bout, including both movement and stationary periods. | Wheels |
| Movement duration* | Total time spent in motion during a mobility bout. | Wheels |
| Continuity* | Ratio of time spent moving to the total duration of a mobility bout. | Wheels |
| Mean speed* | Average speed during the time in motion within a mobility bout. | Wheels |
| Max speed* | Highest speed reached during the time in motion within a mobility bout. | Wheels |
| Distance* | Total distance covered during a mobility bout. | Wheels |
| Displacement* | Straight-line distance between the starting and ending points of a mobility bout. | WC frame |
| Mean slope* | Average incline or decline encountered during a mobility bout. This measure helps distinguish between actual slope navigation during movement and simply leaning backward or forward while at rest. | WC frame |
| ADA compliance* | Number of slopes encountered during a mobility bout that exceed the American Disability Act recommendation (4.76 degrees or 1:12 ratio). | WC frame |

* Metric calculated for each mobility bout.

nontraditional schedules or night shifts. This approach prevents splitting periods of continuous activity and provides a more accurate representation of daily arm loading.

- **Number of turns:** The number of turns was determined using the IMU attached to the wheelchair frame relative to the inertial frame. We defined a turn when the angular velocity exceeded a threshold of 10°/sec, where short pauses (≤ 0.05 seconds) between movements were merged into ongoing movements and movement bouts shorter than 1 second were excluded. These thresholds were informed through the analyses of pilot data measured during an in-lab data collection and a supervised outdoor data collection; turns were confirmed by examining the MWC position trajectories, which included figure-eight routes and outdoor propulsion on a college campus.

- **Number of starts and stops:** Starts and stops were identified when wheelchair speed exceeded ±0.1 m/s for at least 2 seconds. A 1-second drop in wheelchair speed was allowed within each event, and at least one wheel revolution was required to count as a start or stop.

- **Mobility bouts:** Mobility bouts began when wheelchair speed exceeded 0.1 m/s with at least one complete wheel revolution and continued through intermittent stops until there was one minute of continuous inactivity (Fig 5). This definition captures the stop-and-start nature of free-living wheelchair propulsion, with the one-minute threshold based on average traffic light cycle times adapted from the definition of the maximum resting period from [31] and the average cycle time of traffic lights (National Association of City Transportation).

- **Continuity:** Ratio of time spent moving to the total duration of a mobility bout, with 100% being representative of a continuous mobility bout with no stopping during propulsion [23].

$$\text{Continuity} = \frac{\text{total duration} - \text{ no movement duration}}{\text{total duration}} * 100\% \tag{5}$$

- **Total distance**: Total distance per mobility bout was calculated using the number of complete wheel revolutions, calculated from wheel radius and wheelchair speed [32].

- **Displacement:** The total change in position of the wheelchair from the starting location of the bout, calculated using the trajectory derived from the yaw angle ($\theta_{yaw}$) and wheelchair linear velocity. The yaw angle was calculated from the orientation of the wheelchair axis $\hat{X}_{WC}$ relative to the inertial frame:

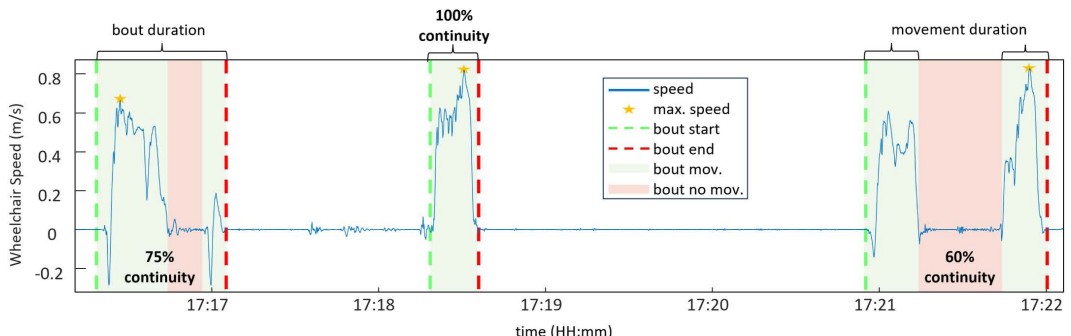

**Fig 5. Movement bouts and bout continuity.** Time series plot showing wheelchair speed data for a single participant, illustrating key bout metrics. The figure highlights three distinct movement bouts.

$$\theta_{yaw} = \tan^{-1}\left(\frac{R_{WC\_inertial\|IMU\_WC}(2,1)}{R_{WC\_inertial\|IMU\_WC}(1,1)}\right)$$

(6)

where $R_{inertial\|WC}$ represents the orientation of the inertial reference frame relative to the wheelchair.

The velocities of the wheelchair in the x and y directions, $v_x$ and $v_y$, were computed as follows:

$$v_x = v_{WC} * \cos(\theta_{yaw})$$

(7)

$$v_y = v_{WC} * \sin(\theta_{yaw})$$

(8)

These velocities were integrated over time to determine the displacement in the x and y directions, denoted as $d_x$ and $d_y$, respectively. The total displacement was then calculated using the magnitude of the 2D displacement vector, which combines both x and y directional movements to give the overall change in position.

- **Slope:** The slope angle was defined by the pitch angle ($\theta_{slope}$) defined by the vertical orientation relative to the wheelchair-fixed frame

$$\theta_{slope} = \tan^{-1}\left(\frac{R_{WC\_inertial\,|\,IMU\_WC}(3,1)}{R_{WC\_inertial\,|\,IMU\_WC}(3,3)}\right)$$

(9)

A moving average filter using a window size of 5 seconds was used to smooth the data by averaging it over a 500 samples (5 seconds × 100 samples/second). Slope was considered non-zero when the pitch angle exceeded 0.5° and was maintained for at least one meter. Fig 6 illustrates wheelchair speed and slope data for a single participant over a portion of a day.

- **ADA compliance:** The number of slopes encountered during a bout where the slope angle exceeded the American disability act recommendation (4.76 degrees or 1:12 ratio).

### Statistical analyses

MWC mobility bouts were analyzed using five metrics (duration, continuity, mean speed, total distance, and displacement data, chosen based on prior studies of WC mobility and were selected to provide a comprehensive view of both temporal and spatial aspects of movement) from twelve participants. A k-means cluster analysis was performed to categorize movement bouts. This method enabled grouping bouts with similar movement characteristics across multiple dimensions, revealing distinct patterns of mobility. The number of clusters was chosen using the elbow method [33], which determined that 3 clusters was the ideal number. Given the exploratory nature of this analysis, clusters were identified to describe natural groupings within the data, without conducting formal statistical comparisons between groups.

Descriptive statistical analysis was conducted on 7 days of mobility data from 12 manual wheelchair users, categorized by injury level (Mid: T1-T8, n = 6; High: C6-C8, n = 3; Low: T9-L1, n = 3). For each participant, means, standard deviations, medians, and ranges were calculated across the 11 mobility metrics. Within-person and between-person variability was assessed visually by examining data distribution patterns, histograms and quantitatively with calculated range and standard deviation. The participants were divided into three groups based on injury level to identify potential differences in mobility characteristics across injury levels.

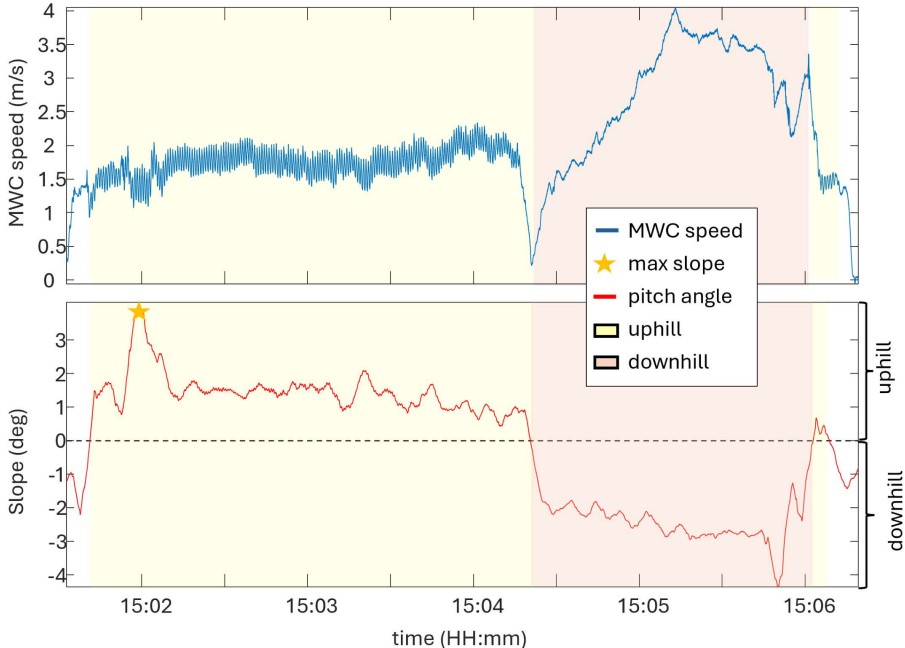

**Fig 6. Relationship between wheelchair speed and slope during real-world mobility.** Wheelchair speed and pitch angle data (slope) for a single participant over a portion of a day (approximately 5 minutes).

## Mobility profiles

Individual mobility profiles were created by calculating totals per week of the number of turns, starts and stops, and slopes steeper than 1:12 rise (ADA compliance slope). Mobility profiles also included the per week number of short duration (<215s), medium duration (215s - 700s) and long duration (>700s) bouts; thresholds for each bout were determined from the k-means cluster analysis. Bout duration thresholds (215 s and 700 s) were derived by examining where natural breaks occurred between clusters across participants, allowing for classification of short, intermediate, and long-duration bouts based on real patterns observed in the data. Since short mobility bouts account for 94% of all bouts, additional metrics were calculated for short mobility bouts, including total movement duration, total distance covered, mean velocity, and maximum velocity attained.

The mobility profiles provide insight into the physical demands posed by the MWC users 'environments. Together, these metrics offer a comprehensive assessment of participants' movement patterns, capturing the dynamic aspects of their mobility and their interaction with varying terrain.

## Results

### Mobility bouts

Our analysis incorporates data from all participants (n = 12) collected over a total of 84 days, encompassing a total of 1,213 hours of participant data analysis time and 6,024 mobility bouts. On average, participants recorded 16 ± 2 hours (median: 16 hours; range: 9–22 hours) of data analysis time and 502 ± 109 mobility bouts per day (median: 494; range: 328–674). These mobility bouts included a cumulative movement duration of 92 hours and a total distance of 125 km for all participants. The median mobility bout lasted 40 seconds, covering 9 meters at an average speed of 0.43 m/s (Table 3). The substantial differences between mean and median values highlight a skewed distribution, with outliers contributing to consistently higher means than medians, underscoring the impact of long duration mobility bouts on the overall statistical distribution.

**Table 3. Descriptive statistics for all mobility bouts (n = 6024).**

| Mobility bout metrics | Mean ± std | Median (range) |
|---|---|---|
| Duration (s) | 72 ± 114 | 40 (3-3720) |
| Movement duration (s) | 55 ± 95 | 33 (3-3672) |
| Distance (m) | 21 ± 102 | 9 (0-5287) |
| Mean speed (m/s) | 0.43 ± 0.16 | 0.40 (0.09-1.48) |

Traditional mobility metrics showed patterns of variability that differed between and within participants (Fig 7). Within-participant variability appears more pronounced than between-participant variability across mobility bouts movement duration, mean speed, and total distance. Participants were categorized into Low (T9-T1), Mid (T1-T8), and High (C6-C8) level of SCI groups, however, no clear patterns emerged for the measured bout metrics across these groups. Movement durations were 24.9–43.64 seconds, with Participant 1 (male, age range = 30–39 years, time since injury = 9 years, SCI level = Mid) showing the highest median duration and largest variability. Mobility bout total distances showed high within-person variability, with median values between 6.94–15.08 meters, although Participants 1 and 6 (male, age range = 50–59 years, time since injury = 28 years, SCI level = Mid) covered notably longer distances. Median bout speeds ranged from 0.34–0.48 m/s, with Participant 6 showing the highest median speed and Participant 8 (male, age range = 30–39 years, time since injury = 12 years, SCI level = High) the lowest for all mobility bouts.

To better understand the range of movement speeds for each participant (between-day, within-person), we supplemented the mean speed data with a histogram illustrating the distribution of raw MWC speeds recorded over a week (Fig 8). This approach captures the variability in the speed ranges, emphasizing that individuals exhibit diverse speed profiles across movement bouts.

The different MWC speed distributions and time spent at each speed highlights the varying mobility patterns between participants, which can be further understood by examining the characteristics of different mobility bouts (Fig 9). For example, short bouts had high continuity (87%; defined as the ratio of time spent moving to the total duration of a mobility bout), covered a mean displacement of 5.91 m, and had an average mean speed of 0.42 m/s. Medium duration bouts were less continuous (69%), had a mean displacement of 26.83 m, and a mean speed of 0.49 m/s. Long duration bouts had high continuity (85%), covered a mean displacement of 186.98 m, and had a mean speed of 0.75 m/s. Shorter-duration bouts accounted for almost 94% of the total bouts compared to long bouts, which accounted for only 0.24% of the bouts during the 7-day collections.

High-resolution mobility metrics characterized each participant's mobility, including the number of bout types, turns, starts and stops, and the number of slopes exceeding the ADA recommend 1:12 ratio (Table 4). Participant 6, with a mid-level injury, demonstrated the highest overall activity, recording 10,290 turns, 4273 starts/stops, and 674 mobility bouts across 7 days. In contrast, Participant 12, with a low-level injury, showed the least activity with 3,377 turns, 1,426 starts/stops, and 328 mobility bouts.

Table 5 summarizes key mobility metric daily statistics for each participant, including movement duration, distance traveled, and MWC speed. It also provides more detailed insights into the complexity of daily wheelchair use, highlighting the frequency of turns, starts and stops, encounters with steep slopes, and the number of mobility bouts.

We analyzed the mean bout slopes across all participants showing the distribution of mean bout slopes for each of the 12 participants (Fig 10), which revealed significant variability in the terrain navigated by individuals with SCI. Most bouts for all participants occurred on relatively flat surfaces, with mean slopes clustering between 0 and 2 degrees. However, each participant demonstrated the ability to navigate steeper inclines, as evidenced by the scattered data points at higher angles. Some participants showed instances of navigating slopes exceeding the ADA recommendation.

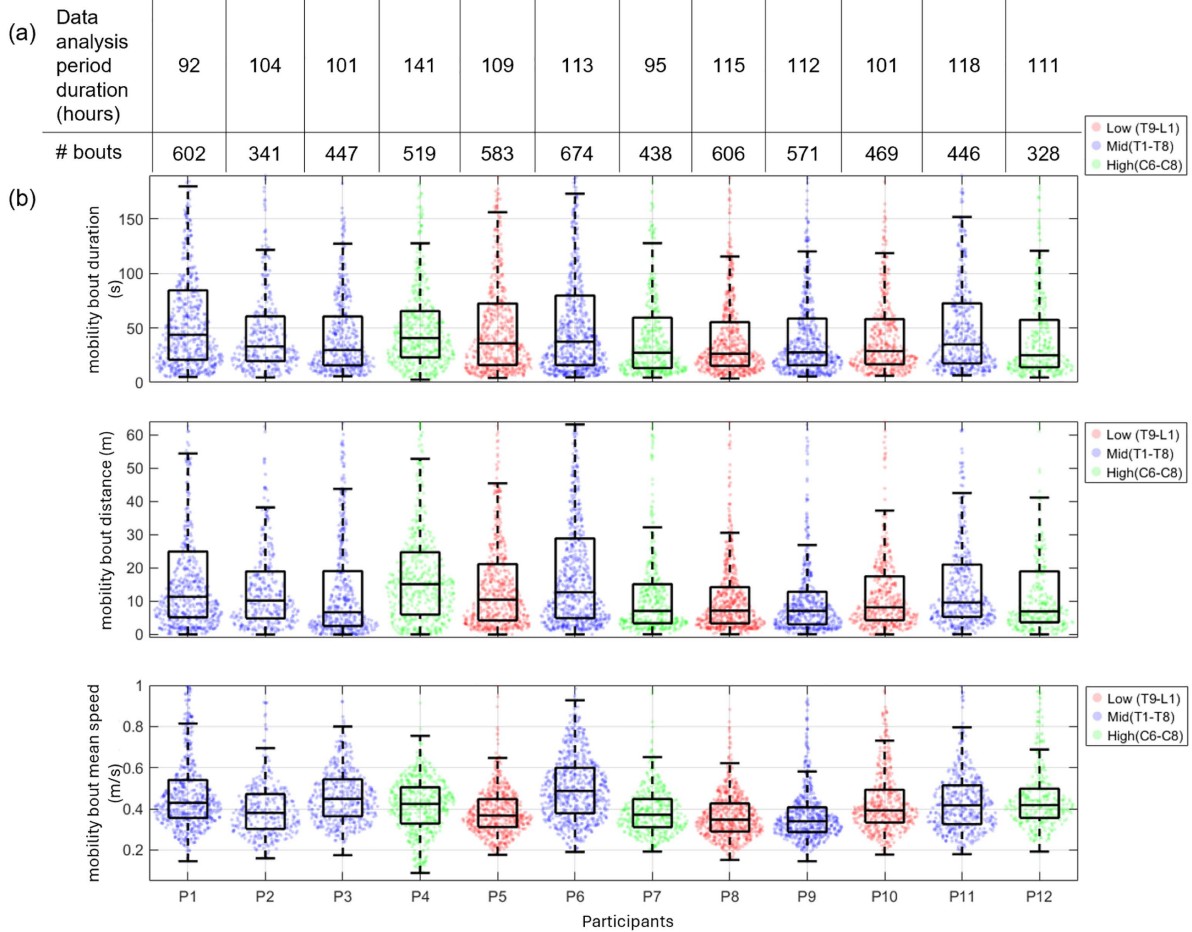

**Fig 7. Comparison of mobility metrics across participants and SCI levels. (a)** Total data analysis periods in hours and number of movement bouts per 7-day collection by participant. **(b)** Mobility bout movement duration **(s)**, distance **(m)**, and mean speed (m/s) across participants. Participants with red data points have a low-level SCI, blue data points have a mid-level SCI, and green data points have a high-level SCI.

Short-duration bouts accounted for 5675 out of 6024 total bouts for all participants. A comparison of short-duration mobility bout metrics grouping participants with Low (T9-L1), Mid (T1-T8), and High (C6-C8) levels of SCI is depicted in Fig 11. The spider plots illustrate each participant's number of bouts in 7 days, total 7-day movement duration, mean speed, max speed, and total distance covered in 7 days during all short duration bouts. Participants exhibited diverse data profiles with notable individual variations, and no clear pattern emerged due to spinal cord injury (SCI) level classification. Each participant demonstrated unique mobility characteristics, with some individuals showing significantly higher values across multiple parameters.

## Mobility profile

We created a MWC user mobility profile for each participant. The profile is a summary of 7-day mobility metrics and includes the number of turns, starts/stops, steep slopes, and bouts. For short duration bouts. A spider plot with 7-day movement duration, mean speed, maximum speed, 7-day distance, and the number of short duration bouts was created. Finally, the mobility profile includes a scatter plot of slopes navigated during 7 days of data collection with an ADA

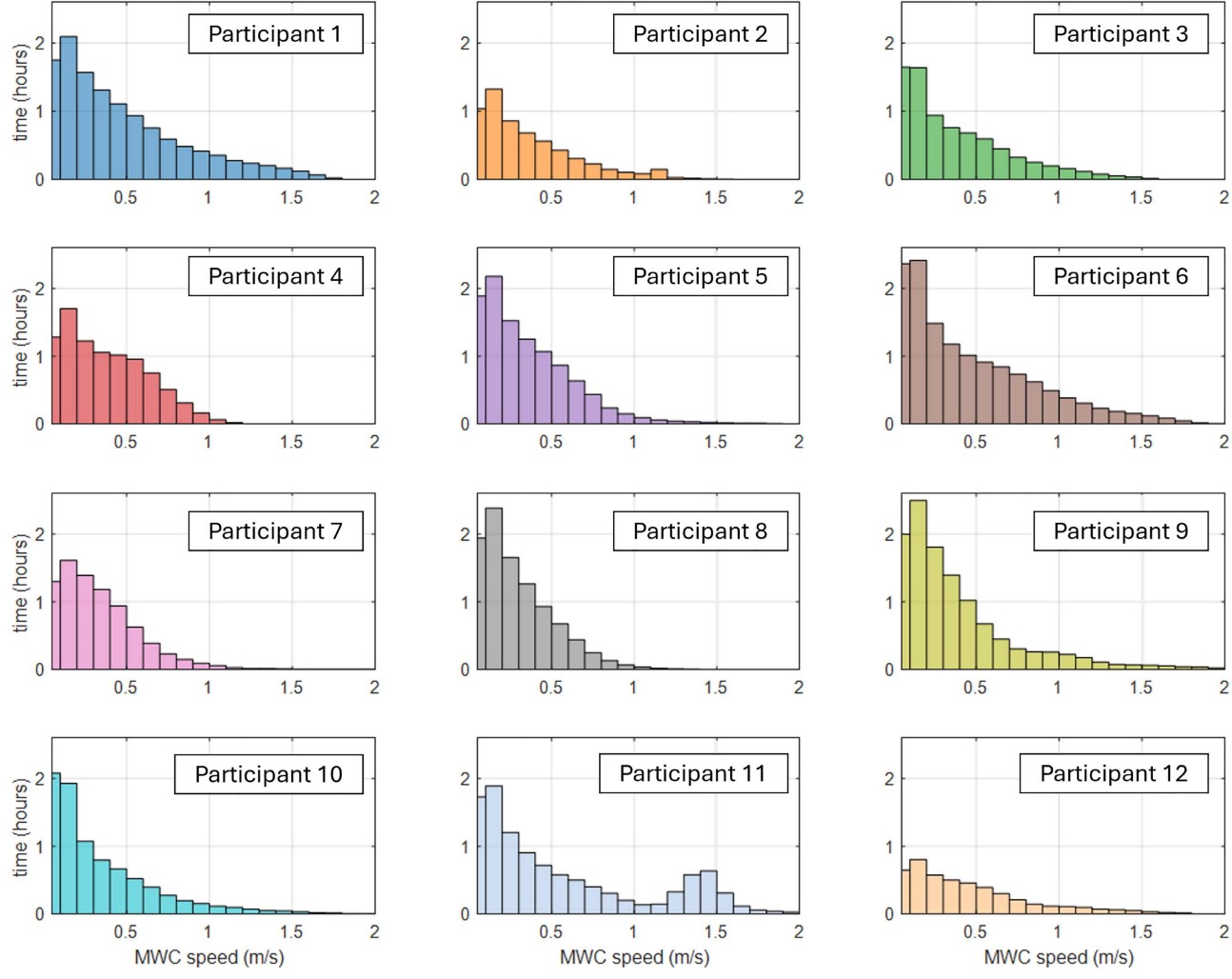

**Fig 8. Distribution of time spent at different wheelchair speeds for each participant over one week.** The bin width in the histogram is 0.1 m/s. Note that only the speeds between 0 and 2 m/s are shown to provide a closer view of the slower-speed range.

recommendation line drawn at 4.8 degrees. Fig 12 compares the mobility profiles of the most active (P6) and least active (P12) participants.

## Discussion

Quantifying demanding physical movements such as the number of turns, number of starts and stops, and slopes that exceed the ADA compliance is crucial for understanding mobility demands for the individual and comparisons between participants. Our integrative approach to mobility profiling represents a novel contribution to wheelchair research by combining traditional measures (speed, distance, and mobility bouts) with previously unexamined metrics including start/stop frequency and ADA slope violations. These high-resolution mobility metrics provide critical context to conventional measures, revealing the physical demands and environmental challenges that measures of speed and distance alone cannot capture. The quantification of ADA slope violations is novel in wheelchair mobility research and offers insights into environmental barriers encountered during daily life.

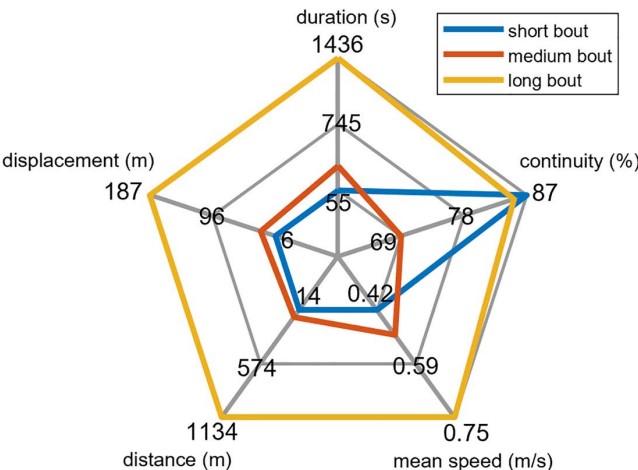

**Fig 9. Mobility bout classification.** Comparison of the mean mobility bout characteristics for short (blue), medium (orange), and long (yellow) duration bouts for all participants.

**Table 4. Mobility metrics for 7-day collection.**

| Participant | Injury level | turns | starts/stops | Slopes> (1:12) | Short duration bouts (<215s) | | Medium duration bouts (215s < duration < 700s) | | Long duration bouts (> 700 s) | |
|---|---|---|---|---|---|---|---|---|---|---|
| | | | | | No. bouts | distance (m) | No. bouts | distance (m) | No. bouts | distance (m) |
| P1 | Mid | 7774 | 3505 | 1 | 548 | 10078 | 52 | 5401 | 2 | 936 |
| P2 | Mid | 4618 | 2084 | 2 | 329 | 4563 | 12 | 1283 | 0 | 0 |
| P3 | Mid | 5088 | 2148 | 0 | 430 | 5429 | 17 | 1267 | 0 | 0 |
| P4 | Low | 6908 | 3240 | 0 | 485 | 7530 | 34 | 2040 | 0 | 0 |
| P5 | High | 8139 | 3759 | 1 | 545 | 7531 | 36 | 2204 | 2 | 286 |
| P6 | Mid | 10290 | 4273 | 0 | 608 | 10124 | 63 | 4929 | 3 | 2634 |
| P7 | Low | 4922 | 2519 | 3 | 416 | 4787 | 18 | 1100 | 4 | 1377 |
| P8 | High | 7172 | 3723 | 0 | 584 | 5601 | 22 | 1002 | 0 | 0 |
| P9 | Mid | 7292 | 3958 | 1 | 546 | 6062 | 24 | 5043 | 1 | 1196 |
| P10 | High | 5353 | 2755 | 8 | 449 | 5793 | 20 | 1644 | 0 | 0 |
| P11 | Mid | 5468 | 2550 | 10 | 417 | 6032 | 26 | 2810 | 3 | 10576 |
| P12 | Low | 3377 | 1426 | 0 | 318 | 5214 | 10 | 534 | 0 | 0 |
| all | | **76401** | **35940** | **26** | **5675** | **78745** | **334** | **29256** | **15** | **17004** |

Our study, conducted with 12 participants over a total of 84 days (7 days per participant), provides insights into wheelchair mobility during daily life and provides a blueprint for adding high-resolution mobility metrics to research and clinical settings. While our sample size is smaller than some previous studies, our findings corroborate and diverge from existing literature in interesting ways. The data suggest the benefit of individualized assessments and interventions for people with SCI. Table 6 compares our findings to the results of five other studies [11,12,14–16] on MWC mobility in adults. These studies were chosen for comparison because they similarly focused on manual wheelchair mobility in adults with SCI, used comparable metrics, and collected data over extended durations, making their findings relevant and comparable to ours. Despite differences in study populations, daily distance outcomes are generally similar, except for one study on competitive athletes [15] and another conducted in a different country [14]. These discrepancies likely stem from multiple

**Table 5. Key daily mobility metric statistics.**

|  | Mean±std | Median (range) |
|---|---|---|
| Movement duration (min) | 65.44±21.81 | 65.26 (32.66-103.76) |
| Distance (m) | 1488±700 | 1215 (821-2774) |
| Mean speed (m/s) | 0.43±0.04 | 0.42 (0.36-0.51) |
| No. bouts | 72±16 | 71 (47-96) |
| No. starts/stops | 428±125 | 428 (204-604) |
| No. turns | 910±272 | 884 (204-610) |
| No. slopes> (1:12) | 4±5 | 2 (0-14) |

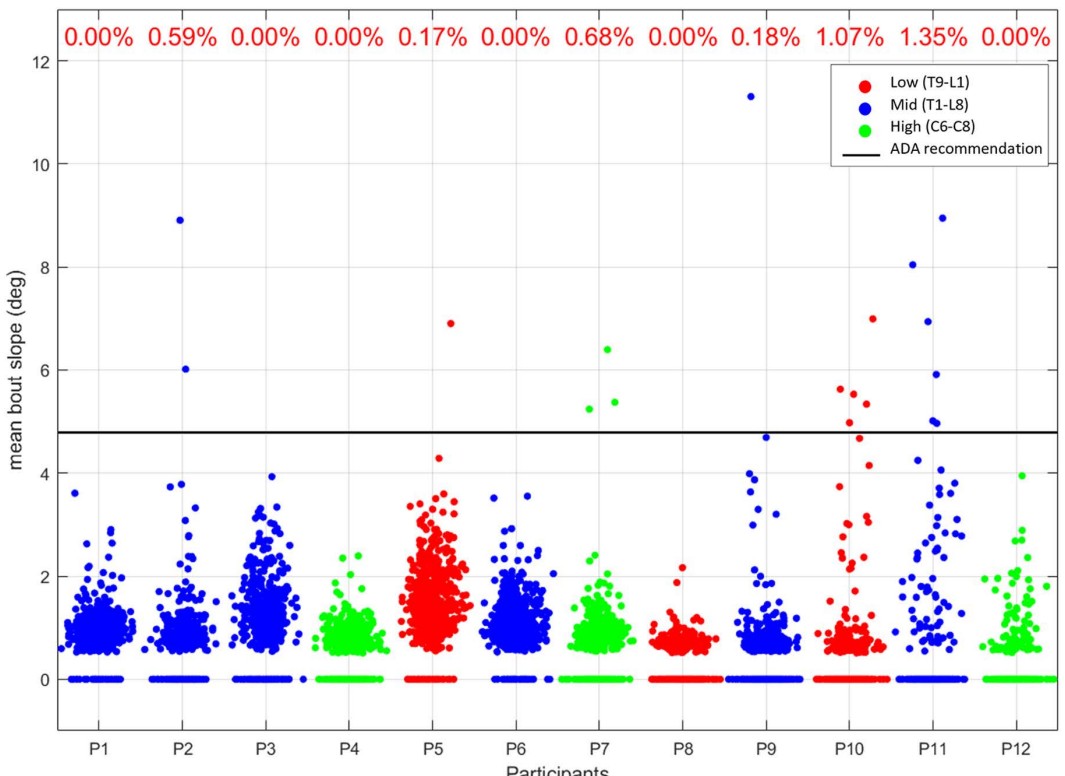

**Fig 10. Mean bout slopes across all movement bouts for 12 participants.** The x-axis represents the participants, while the y-axis shows the mean bout slope in degrees, with data ranging from 0 to 11 degrees. The black line at approximately 4.8 degrees represents the ADA (Americans with Disabilities Act) compliant slope angle limit, and the percentage represents the percentage of each participant's navigated slopes that were steeper than the ADA recommendation.

factors, though the relative contribution of each cannot be definitively established from the available data. Variations in sample sizes, community settings, and participant demographics represent plausible explanatory factors. The higher distances observed in the, the National Veterans Wheelchair Games (NVWG) could reflect the more active lifestyle characteristics typical of competitive athletes, as competitive sport participation has been associated with higher physical activity levels in other disabled populations. However, other unmeasured factors such as differences in measurement protocols, environmental contexts, or participant motivation may have also influenced these findings. Similarly, while previous research has documented associations between employment status and greater wheeled distances [14,15], the extent

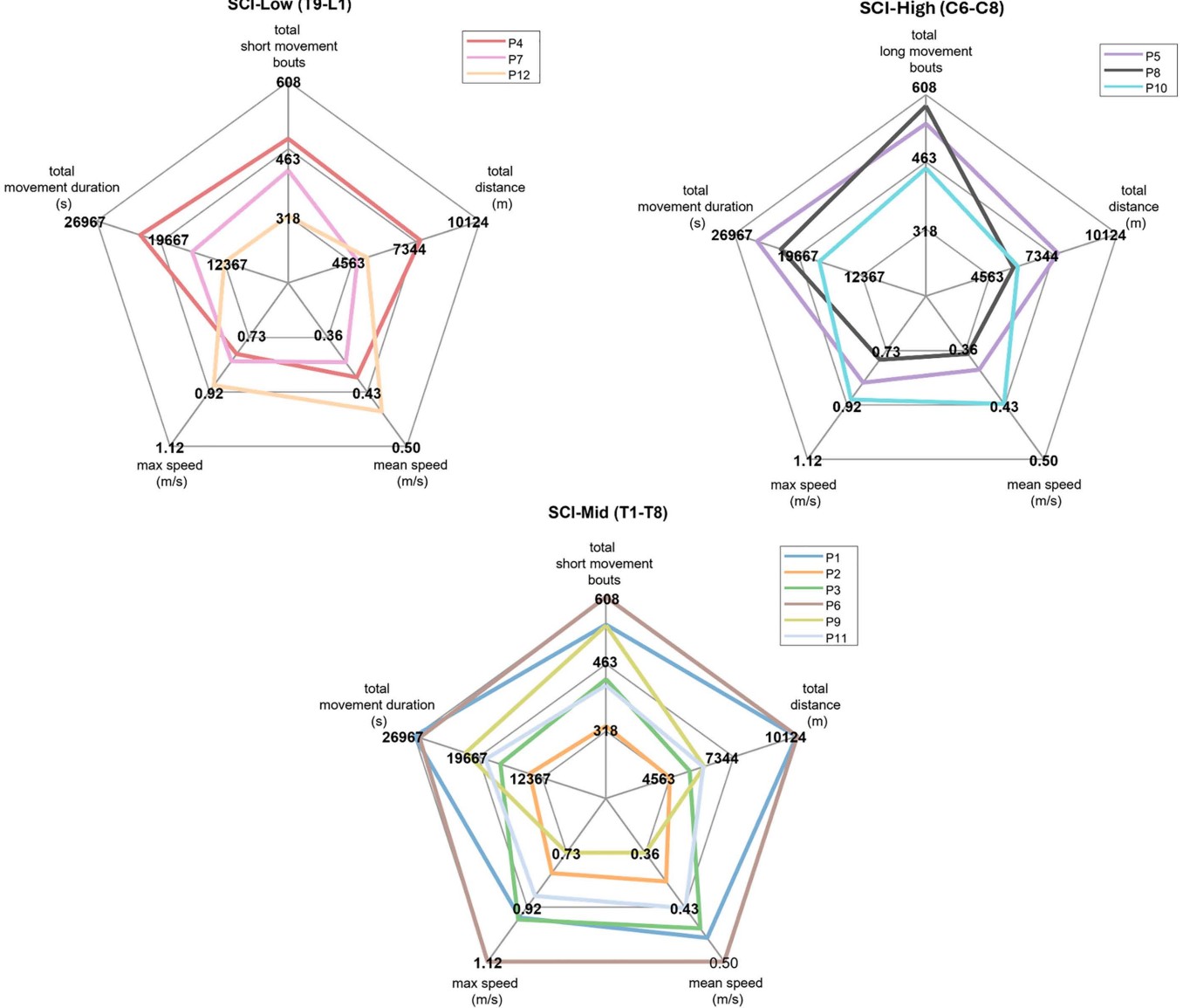

**Fig 11. 7-day mobility metrics across three SCI injury levels: Low (T9-L1), Mid (T1-T8), and High (C6-C8).** Each plot's scale starts at zero in the center, with the maximum value representing the highest value across all subjects for all short duration movement bouts. Total distance and movement duration represent the cumulative 7-day data for all short movement bouts.

to which employment or other socioeconomic factors explain cross-study differences remains uncertain. Geographic and cultural differences between studies may also contribute, but identifying the specific mechanisms linking country of origin to mobility patterns would require further investigation. Future research should prioritize standardized measurement protocols and more comprehensive demographic characterization to enable more definitive conclusions about factors influencing daily mobility patterns in wheelchair users.

Our analysis of bout lengths revealed some differences compared to previous research. We found longer mean bout lengths than previously reported [12]. While these differences can be partially attributed to variations in mobility bout definitions across studies, they may also reflect genuine behavioral and contextual differences in mobility patterns.

| Metrics | P6 | P12 |
|---|---|---|
| Age (years) | 50-59 | 50-59 |
| Sex | Male | Female |
| Time since injury (years) | 29 | 10 |
| Injury level | Mid (T1-T8) | Low (T9-L1) |
| # turns | 10290 | 3377 |
| # starts/stops | 4273 | 1426 |
| # slopes > (1:12) | 0 | 0 |
| # short bouts | 608 | 318 |
| # medium bouts | 63 | 10 |
| # long bouts | 3 | 0 |

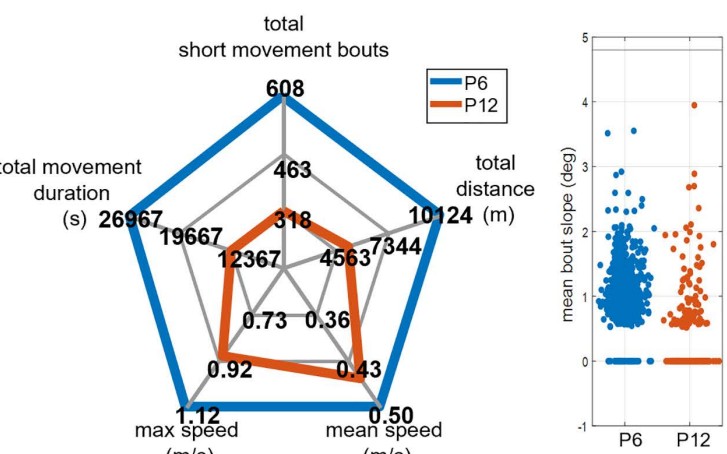

**Fig 12. Comparison of 7 Day Mobility Profiles between Participant 6 (P6) and Participant 12 (P12), across various metrics.** P6 (blue) and P12 (orange), are both 50-59 years old. The metrics show P6 had more turns (10290 vs. 3377), starts/stops (4273 vs. 1426), short duration (<215s) mobility bouts (608 vs. 318), and medium (215-700s) and long duration (>700s) bouts. The short duration bout spider plot also reveals P6 had longer total 7-day movement duration, higher maximum speed, and higher mean speed compared to P12.

Methodologically, our study defined a bout as a period of wheelchair propulsion capturing intentional transitions between activities with pauses less than one minute (beginning when wheelchair speed exceeded 0.1 m/s with at least one complete wheel revolution), previous research used different criteria: bouts lasting at least 5 seconds, with speeds ≥ 0.12 m/s, ending when less than 0.76 m were wheeled within 15 seconds [12]. These different definitions likely account for the observed variations, although they might also reflect genuine differences in mobility patterns influenced by community environments.

However, the longer bout lengths in our study may also indicate true differences in mobility behavior and environmental context. Our participants' community environments may have promoted longer, more sustained periods of wheelchair propulsion compared to previous studies. Factors such as greater distances between destinations, more accessible infrastructure encouraging continuous movement, or different activity patterns within the community could contribute to these extended mobility bouts. Additionally, our participants may have engaged in different types of activities or had mobility goals that naturally resulted in longer propulsion periods.

We defined three distinct categories of movement bouts based on duration, with short bouts (<215s) comprising nearly 94% of weekly activity. The high standard deviations in our bout characteristics also suggest considerable variability in individual mobility, which may reflect both personal preferences and diverse environmental demands within the community setting.

Each participant shows considerable spread in their measurements (Fig 7), as evidenced by the scattered data points extending well beyond their respective box plots and numerous outliers. In contrast, between-participant variability appears more modest, with median values remaining relatively consistent across participants and is particularly visible in mean speed where medians cluster around 0.4 m/s. The interquartile ranges (boxes) also show similar patterns across participants, suggesting that while individuals exhibit high variability in the higher levels of mobility metrics, the central tendencies are fairly consistent across different participants.

The histograms (Fig 8) highlight that higher speeds occur less frequently, indicating that while average speeds provide a useful measure, the distribution offers useful insight into individual mobility, illustrating how often participants reach different speeds within the week. The majority of mobility time is spent at lower speeds, with a noticeable decrease in time

**Table 6. Comparison of studies measuring daily MWC use in everyday life.**

| Study | Collection Setting | Equipment | Subject Characteristics | Distance (km) | Speed (m/s) | Movement Time (min) | Total Bouts | Number of Turns |
|---|---|---|---|---|---|---|---|---|
| This study | In-home and community | IMU | Adults<br>years of age: 47 ± 13.4<br>SCI (n = 12) | 1.5 ± 0.7 | 0.43 ± 0.04 | 65.5 ± 21.8 | | 910 ± 272 |
| **Tolerico et al. [15]** | In-home and community | Data logger* | Athletes<br>years of age: 46.8 ± 13.3<br>SCI (n = 40)<br>others (n = 12) | 2.5 ± 1.2 | 0.79 ± 0.19 | 47.9 ± 21.4 | – | – |
| **Oyster et al. [11]** | In-home and community | Data logger* | Adults<br>years of age: 39.37 ± 12.66<br>SCI (n = 132) | 1.9 ± 1.1 | 0.63 ± 0.12 | 46.8 ± 23.6 | – | – |
| **Sonenblum et al. [12]** | In-home and community | Accelerometer | Adults<br>SCI (n = 20)<br>others (n = 8) | 1.9 ± 1.5 | 0.48 | 58.1 ± 37.6 | 96 ± 50<br>(21 seconds,<br>8.6 m) | – |
| **Levy et al. [16]** | In-home and community | Bicycle computers | Adults<br>years of age: 43.1 ± 14.5<br>SCI (n = 14)<br>others (n = 6) | 1.5 ± 1.7 | – | – | – | – |
| **Togni et al. [14]** | In-home and community Switzerland | IMU | Adults<br>years of age: 37 ± 12<br>SCI (n = 12)<br>others (n = 2) | 3.1 ± 1.5 | 0.35 ± 0.05 | 71 ± 24 | – | 913 ± 274 |

*Data logger: The data logger tracks wheelchair wheel rotation using three reed switches spaced 120° apart and a gravity-stabilized magnet. Each 120° rotation triggers a switch, recording a timestamp to the nearest tenth of a second [15]

as speed increases (Fig 8). Tracking these less frequent but higher levels of mobility may be important to understanding the physical demands of navigating environments or certain activities and understanding why some individuals develop mobility-related health problems while others do not.

Our study contributes novel new information on continuity, the frequency of starts and stops (428 ± 125 per day), and encounters with slopes exceeding ADA recommendations (2.72 ± 3.35, per week), providing valuable insight into the physical demands that wheelchair users face in navigating their environments. In addition to the high physical demands of movement, more force and torque are required when MWC users propel on surfaces with greater resistance, such as ramps, interlocking pavers, and grass [34,35]. While most mobility occurred on relatively flat surfaces, participants occasionally navigated slopes exceeding ADA-compliant limits. Our study also underscores the importance of analyzing the number of starts, stops, and turns, as these actions require larger changes in linear and angular momentum compared to steady-state propulsion [36]; previous research has demonstrated that stress on the upper extremities during propulsion is higher during acceleration than during constant velocity. Additionally, studies have shown that the physical demands during the starting phase of wheelchair propulsion are comparable to those experienced during ramp propulsion [37], further emphasizing the importance of considering these transitional movements in MWC mobility analyses.

Participants 10 and 11 exhibited the highest concentration of bouts with average slopes near or above the ADA limit, suggesting either a more challenging living environment or greater mobility capabilities. In contrast, participants 4, 8, and 12 showed the least variability in slope navigation, with most bouts occurring on flatter surfaces. While outliers reaching up to 17 degrees were observed for some participants, indicating navigation of significantly steep inclines, these steeper sections may have included instances of assisted pushing. Future work will incorporate arm movement sensors to definitively distinguish between self-propulsion and assistance during these high-slope regions.

We observed substantial variability in mobility patterns across 12 SCI subjects over a week, leading to the development of an SCI mobility profile that incorporates various metrics to characterize individual mobility patterns (Fig 12). While the distance traveled per day, mean speed and minutes of MWC mobility per day results align with existing wheelchair mobility literature, our high-resolution mobility metrics reveal significant individual variability among participants with SCI, suggesting that understanding these differences can provide valuable insights into how mobility demands relate to factors like community participation, quality of life, and health, highlighting the importance of personalized assessment in rehabilitation strategies and assistive technology design.

This high intra-individual variability has several important clinical and technological implications. For rehabilitation planning, it suggests that mobility assessments based on single-day or short-term observations may not accurately represent an individual's typical patterns, potentially leading to suboptimal therapeutic targets. Clinicians should consider incorporating week-long monitoring to establish realistic baseline mobility goals and track meaningful changes over time. For device design, this variability indicates that 'one-size-fits-all' wheelchair configurations may be inadequate; incorporating adaptive or customizable features that accommodate daily fluctuations in mobility demands could improve user satisfaction and reduce overuse injuries. In health monitoring strategies, understanding an individual's typical variability range allows for better detection of clinically significant changes that warrant intervention, distinguishing between normal day-to-day fluctuations and concerning mobility declines.

There are important limitations to consider when interpreting the results of our study. Our relatively small sample size of 12 participants, limiting the generalizability of our findings to the broader wheelchair user population. We know that two participants experienced non-typical work weeks (due to training and the holidays). As has been mentioned in other studies [16] it is common for individuals to use more than one wheelchair during the week, however we only mounted sensors on one wheelchair. We know that this impacted one of our participants (P4: male, age range=50–59 years, time since injury=27 years, SCI level=Low) who uses a different wheelchair when mobilizing in the lower level of their house. These limitations may have resulted in underestimation of true mobility patterns and biomechanical loading in some participants, potentially affecting the accuracy of our proposed mobility profiles. The missing wheelchair data for participants using multiple chairs likely led to incomplete capture of daily mobility demands, which could influence the validity of mobility-health relationships derived from our data. Additionally, the inability to account for different surface types means our biomechanical interpretations may not fully represent the loading experienced during real-world mobility, limiting the clinical applicability of these findings for predicting shoulder pathology risk.

The purpose of this study was to introduce new MWC mobility analysis methods using high-resolution mobility metrics and to demonstrate the potential utility of our approach. A larger sample size is needed to provide normative data for these metrics. While our study captured various aspects of wheelchair mobility, environmental factors such as surface types (e.g., interlocking pavers, grass) were not systematically documented; these surfaces require higher forces and torques during propulsion [33,34]. The sensor technology used in this study does not directly measure the surface on which the MWC is rolling, however surveys could be added to our approach to gain more information about biomechanically demanding surfaces. Our methodology may miss slopes lasting less than 5 seconds due to the sampling approach and the use of a moving mean filter to reduce vibration-induced noise. While this could prevent detection of brief steep inclines or declines, it has minimal to no impact on our analysis since we calculate average slope per mobility bout, which smooths short-term fluctuations and reflects overall incline conditions.

Relating detailed mobility and biomechanical data to shoulder pathology in manual wheelchair users is complex, as shoulder problems likely result from a combination of factors. While load from wheelchair use is an important contributor, it is only one piece of the puzzle [38]. Other key contributors include repetitive loading, propulsion technique, transfer mechanics, seating posture, arm use during activities of daily living, injury history, muscle strength and conditioning, as well as individual anatomical, health, and psychosocial factors that influence pain perception and behavior. Our high-resolution mobility data provides objective quantification of one key component (wheelchair propulsion exposure) but

must be integrated with clinical assessments of these other factors to develop a comprehensive understanding of shoulder pathology risk. Future studies should combine our mobility analysis approach with longitudinal clinical evaluations, imaging, and patient-reported outcomes to establish how mobility patterns interact with individual risk factors to influence shoulder health outcomes. Our data on mobility bouts and biomechanical loading provides valuable insight, but integrating these findings with clinical assessments and longitudinal monitoring of shoulder health is essential for understanding the multifactorial nature of shoulder pathology and informing strategies to prevent or mitigate it.

## Conclusion

Our analysis employs a novel approach to characterize manual wheelchair mobility during daily life, which enables a unique understanding of the propulsion-related demands experienced by manual wheelchair users. This study demonstrates that the data recorded by three manual wheelchair-mounted inertial measurement provides rich information that can be used to quantify mobility. While mean bout speed, duration, and total distance were consistent across participants, higher-resolution mobility metrics revealed significant individual variability. Specifically, metrics such as the number of turns, starts and stops, and the number and steepness of slopes varied considerably, suggesting differences in upper extremity loading during real-world manual wheelchair use. Mobility characteristics can differ due to variations in community environments, employment status, and weekly activities at the time of data collection. Our findings underscore the need for personalized assessments to better understand mobility and the potential for tailored interventions to reduce the risk of overuse injuries in manual wheelchair users.

These individualized mobility metrics have direct translational potential for clinical practice and intervention development. For instance, users demonstrating high frequencies of starts/stops or steep slope navigation could be prioritized for targeted strength training programs or fitted with power-assist technologies to reduce upper extremity loading. Rehabilitation protocols could be personalized based on mobility profiles, for example, those with extended continuous propulsion might benefit from endurance conditioning, while users with frequent directional changes could receive specialized training in efficient maneuvering techniques. Additionally, these objective data could inform evidence-based wheelchair prescription by matching device specifications to individual usage patterns and provide quantitative baselines for monitoring functional changes during routine clinical follow-ups. From a broader perspective, this work supports fundamental goals of autonomy and community inclusion by developing precise, individualized approaches to prevent secondary health conditions that could otherwise limit full participation in employment, recreation, and social activities. By reducing the burden of preventable shoulder injuries, this research contributes to public health objectives of maintaining functional independence, reducing healthcare costs, and enabling aging in place for manual wheelchair users.

Future research should focus on developing methods to distinguish between self-propulsion and assisted propulsion in manual wheelchair users; incorporating additional body-worn sensors will enhance the accuracy and reliability of these detections. Additionally, identifying surface types and quantifying transfers are important factors that impact upper extremity loading that need further exploration. Expanding data collection to include a more diverse range of individuals from different communities will be crucial for understanding how mobility and biomechanical demands vary across populations and communities. Longitudinal studies are also needed to track changes in biomechanical demands and shoulder health over time, enabling the development of proactive strategies to prevent shoulder health decline.

Additionally, exploring interventions such as new wheelchair designs, assistive technologies, or training programs could help mitigate the negative impacts of propulsion demands on musculoskeletal health. Integrating additional variables, such as individual physical characteristics and wheelchair configurations, into future analyses could uncover deeper insights into the factors influencing mobility profiles and lead to more personalized recommendations. Addressing these areas will build on our current knowledge and contribute to improving the quality of life for manual wheelchair users.

## Supporting information

**S1 Data. Dataset of movement bout metrics for all participants (n = 12).**
(CSV)

**S2 Data. Dataset of daily movement metrics: turns, bouts, starts, and stops.**
(XLSX)

**S1 Appendix. Steps to synchronize wheelchair mounted sensors.**
(PDF)

## Author contributions

**Conceptualization:** Kathylee Pinnock Branford, Stephen M. Cain.

**Formal analysis:** Kathylee Pinnock Branford, Stephen M. Cain.

**Methodology:** Kathylee Pinnock Branford, Meegan G. Van Straaten, Omid Jahanian, Melissa M. B. Morrow, Stephen M. Cain.

**Project administration:** Melissa M. B. Morrow, Stephen M. Cain.

**Software:** Kathylee Pinnock Branford, Stephen M. Cain.

**Supervision:** Melissa M. B. Morrow, Stephen M. Cain.

**Visualization:** Kathylee Pinnock Branford.

**Writing – original draft:** Kathylee Pinnock Branford.

**Writing – review & editing:** Kathylee Pinnock Branford, Meegan G. Van Straaten, Omid Jahanian, Melissa M. B. Morrow, Stephen M. Cain.

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
