## [Decision Letter · Decision Letter 0]

30 May 2025

PONE-D-25-16850An inertial sensor-based comprehensive analysis of manual wheelchair user mobility during daily life in people with SCIPLOS ONE

Dear Dr. Cain,

Thank you for submitting your manuscript to PLOS ONE. After careful consideration, we feel that it has merit but does not fully meet PLOS ONE’s publication criteria as it currently stands. Therefore, we invite you to submit a revised version of the manuscript that addresses the points raised during the review process.

 Overall, the reviewers thought this was a strong manuscript but there are several things that need to be clarified or revised.

We look forward to receiving your revised manuscript.

Kind regards,

Anne E. Martin

Academic Editor

PLOS ONE

 [This publication was made possible by funding from the National Institutes of Health (NIH; grant no. R01HD84423). NIH had no role in study design, data collection and analysis, decision to publish, or preparation of the manuscript.]. 

4. Please ensure that you refer to Figure 6 in your text as, if accepted, production will need this reference to link the reader to the figure.

Additional Editor Comments (if provided):

Reviewers' comments:

Reviewer's Responses to Questions

**Comments to the Author**

1. Is the manuscript technically sound, and do the data support the conclusions?

Reviewer #1: Yes

Reviewer #2: Yes

2. Has the statistical analysis been performed appropriately and rigorously? 

Reviewer #1: N/A

Reviewer #2: Yes

3. Have the authors made all data underlying the findings in their manuscript fully available?

Reviewer #1: Yes

Reviewer #2: Yes

4. Is the manuscript presented in an intelligible fashion and written in standard English?

Reviewer #1: Yes

Reviewer #2: Yes

5. Review Comments to the Author

Reviewer #1: An inertial sensor-based comprehensive analysis of manual wheelchair user mobility

during daily life in people with SCI

General:

Carefully written manuscript on wheelchair user mobility monitoring with IMUs.

Reference 10 is not on wheelchair mobility but shoulder load, probably a mix-up with another publication of the author on wheelchair mobility metrics

Methods:

line 109: when data is not used, there is no need to describe the sesnors on the wheelchair user … saves you some words

line 111: recoding? or recording ?

line 132: I do understand wat you do for synchronization, but your description is far from complete. First, you should describe how your sensor is aligned with gravity, which is per definition impossible for the wheelsensor, as it will turn around the wheelaxis during the pivoting movements. Just mentioning "a vertical axis" is not sufficient here. Furthermore, a vertical axis refers to a real world coordinate system, where inertial reference frame refers to a local (sensor) reference frame. Please be as concise as possible.

line 138: please describe in more detail how you calculate orientation quaternions from angular velocity and accelerations.

Furthermore, at line 148-149 you suddenly talk about direction cosine matrix. please be consistent in your wording.

line 152-153: nice approach :-)!!

line 157-158: hope you multiplied by 2*PI as well ??? A simple formula would be helpful here to remove any doubt. Was any filtering applied before numerically differentiating into wheelchair linear acceleration??

line 169-171: you also assume or checked/forced the wheelchair is standing on level ground for this operation?

line 172: you controlled that by placing and tightening the Frame sensor on the bar that connects to both wheel axes, which is per definition, by design, perpendicular wit with frontal axis.

Figure 4. A is a right-handed coordinate system, B and C are left handed coordinate systems.

Although for the metrics derived it doesn't matter too much (left and right turns might be exchanged, and upward or downward slopes), you should adapt your algorithms to it, or how you corrected for this, or describe how it could influence your outcomes (as you don't show any awareness of the difference currently)

line 186: why account for individuals who remain active past midnight. That activity just belongs to the next day. Otherwise, you should adapt your manuscript for wording, and talk about active period in between sleep periods, which sound quite silly.

line 189 – 195: any references for these cutoff numbers, or just decided in discussion based on pilot data analysis? this phrasing is still not informative on how you came to these numbers. Any graphs supporting the choice would be beneficial.

table 2: Slopes. Good one that you take slope during a mobility bout. This is the way to distinguish between slopes, and just leaning backward "at rest". Please discuss shortly, as many readers might not be aware of the distinction, when applying these methods.

line 213: I don't get this. displacement (change in position) is calculated from yaw angle and wheelchair velocity??? Why not just numerical integration of linear velocity? do you expect the wheelchair to move sidewards? Is it the orientation of the inertial frame with respect to the wheelchair, or the wheelchair wrt to the inertial frame (as the latter, the inertial frame, is global)

line 223: so negotiating slopes with a duration of shorter than 5 seconds could be missed. Please elaborate on this in the discussion.

line 231: why did you use a k-means cluster analysis, is a simple histogram not enough. How many groups were chosen for the clustering?

line 238: And? was there any statistical difference between groups defined?

Discussion:

Table 6: Why this selection of studies for comparison? As there are many more, with the same metrics !!

General question: can you/did you distinguish manual wheelchair propulsion from any form of power assisted wheelchair ambulation? If not, how could any unmarked power assisted wheelchair ambulation influence your results?

line 414: Do you have plans to develop methodology to identify surface type from IMU data? Should be partly feasible, when analyzing wheelchair frame sensor data, as higher frequency oscillations on the vertical axis might correlate with pavement type …

And then, the final big question: How to relate all these detailed information to shoulder problems?

And: is the load of wheelchair ambulation the one and only factor that influences progress of shoulder problems (pain, pathology) in manual wheelchair users???

Reviewer #2: Review notes

This study is a valuable addition to gaining insight into the mobility profiles of wheelchair users. With increasing emphasis on the importance of physical activity, information about activity in wheelchair users is still lagging. So, compliments for taking on this challenge. Although the numbers are not that large yet, this is a good first step toward gaining more insight.

Abstract

The abstract is clear but could be more powerful if two points are added. Can you include information about the reliability/accuracy of the method used? Also, some numbers about activity are provided, but how do they relate to previous research? Or, if such comparisons are not available, how do they compare to the able-bodied population (e.g., active time)?

Introduction

Overall, it's comprehensive, well-cited, and logically structured. It provides a clear rationale for the study and situates it within relevant literature.

There is a strong emphasis on both clinical relevance (e.g., secondary health risks) and technological innovation (e.g., IMUs). Prior work is well-referenced, and there is a clear progression from general context to specific gaps in knowledge.

While “mobility profile” is introduced as a novel approach, it’s not clearly defined. Consider clarifying how these profiles are generated or used (e.g., clustering? classification?).

L54–77

This part could be condensed a bit; currently, there is some redundancy.

L80 Could you describe in a bit more detail what is meant by “different environmental contexts”? While it's clear to me, a more general audience might benefit from examples, and from an explanation of how detailed motion measurements might provide more insight.

L82 The shift from discussing the literature to introducing the study could benefit from a clearer, explicit statement. Add something like: “To bridge this gap…”

Methods

Participants

Consider briefly justifying the small sample — is this a pilot study, or a sub-analysis from a larger trial?

Data collection

L109 You state that sensors were placed on the thorax, upper arms, and forearms, but this data was not used. Why was this data collected but excluded from this analysis? If relevant to another study, a short note would clarify.

No mention is made of how data quality was checked — were any sensors dislodged? Was there missing data? Were any days excluded from analysis?

L119 A synchronization movement is used, but since synchronization for the remainder of the day relies on the internal clock of the devices, is once per day enough? Did you analyse drift or variation in sync over the course of a day, and were these consistent across days? If corrections were needed, in what range were they — milliseconds or minutes?

Mobility metrics

This is a clear section, well described. However, the settings and thresholds used are based on in-lab pilot measurements, which could affect their applicability to outdoor or real-world results. It would be good to evaluate how well these settings worked for the collected data. Currently, it's stated that the settings are based on pilot studies, but this lacks rationale. Was this based on visual inspection, trial-and-error, or comparison to ground truth?

For example, why was a 0.05s gap allowed for turning? Why not use a change in direction threshold instead?

Statistical analysis

It is stated that clustering was performed, but it's not explained how k (the number of clusters) was chosen.

You mention “five metrics” (duration, continuity, mean speed, total distance, displacement), but it’s unclear why these specific metrics were selected.

The examination of the data was done “visually” with “histograms and distribution patterns” — it would help to briefly state what specific patterns or anomalies you were looking for.

Mobility profiles

While the thresholds for bout duration were based on k-means, it’s not clear how the numerical cutoffs (215s and 700s) were derived. Since the selection of bout durations directly affects the results and their interpretation, this is an important methodological choice that needs further explanation.

Results

Consider using some subheadings for a clearer structure. Currently, a lot of different analyses are presented together, making it dense to read.

The term “continuity” is used repeatedly (e.g., “short bouts had high continuity”), but it’s never explicitly defined.

Discussion

I am not too enthusiastic about placing Table 6 in the discussion. Usually, this kind of comparison with previous work would belong in the introduction, to show existing gaps and how the new research addresses them.

Emphasize more clearly which specific metrics or analyses are novel (e.g., frequency of starts/stops, ADA slope violations), how they enhance existing methods, and what added insights they provide beyond traditional measures like speed and distance.

The explanation of differences between studies (e.g., athlete vs. non-athlete, different countries) is plausible but speculative and underdeveloped. Either strengthen your arguments with supporting evidence or acknowledge the uncertainty more clearly.

More explicitly discuss whether the observed longer bouts in your study could reflect true behavioural or contextual differences, and not just methodological differences.

The high intra-individual variability is an important finding, but its implications are only lightly addressed. Expand on how such variability could influence rehabilitation planning, device design, or health monitoring strategies.

The limitations are clearly acknowledged, which is good. However, the potential impact of these limitations on specific results or interpretations is not fully articulated. Be more explicit about how these limitations might affect the validity or generalizability of the findings.

Conclusion

While the conclusion notes variability and the need for personalized assessment, it doesn’t fully explain how this could be translated into interventions or clinical practice. Add a line or two about how your metrics could inform practical applications, such as rehabilitation protocols, assistive device design, or long-term health monitoring.

Also, tie your findings back to larger goals: autonomy, inclusion, public health, etc.

6. PLOS authors have the option to publish the peer review history of their article (what does this mean?). If published, this will include your full peer review and any attached files.

Reviewer #1: **Yes: **Wiebe H.K. de Vries

Reviewer #2: No

---

## [Author Response · Author response to Decision Letter 1]

14 Aug 2025

We provide a detailed response to each reviewer and editor comment in the submitted document 'Response to Reviewers.pdf', which can also be found at the end of the pdf build.

---

## [Decision Letter · Decision Letter 1]

9 Sep 2025

An inertial sensor-based comprehensive analysis of manual wheelchair user mobility during daily life in people with SCI

PONE-D-25-16850R1

Dear Dr. Cain,

We’re pleased to inform you that your manuscript has been judged scientifically suitable for publication and will be formally accepted for publication once it meets all outstanding technical requirements.

Kind regards,

Anne E. Martin

Academic Editor

PLOS ONE

Additional Editor Comments (optional):

Reviewer #1:

Reviewer #2:

Reviewers' comments:

Reviewer's Responses to Questions

**Comments to the Author**

1. If the authors have adequately addressed your comments raised in a previous round of review and you feel that this manuscript is now acceptable for publication, you may indicate that here to bypass the “Comments to the Author” section, enter your conflict of interest statement in the “Confidential to Editor” section, and submit your "Accept" recommendation.

Reviewer #1: All comments have been addressed

Reviewer #2: All comments have been addressed

2. Is the manuscript technically sound, and do the data support the conclusions?

Reviewer #1: Yes

Reviewer #2: Yes

3. Has the statistical analysis been performed appropriately and rigorously? 

Reviewer #1: Yes

Reviewer #2: Yes

4. Have the authors made all data underlying the findings in their manuscript fully available?

Reviewer #1: Yes

Reviewer #2: Yes

5. Is the manuscript presented in an intelligible fashion and written in standard English?

Reviewer #1: Yes

Reviewer #2: Yes

6. Review Comments to the Author

Reviewer #1: I do not have any further comments. Comments, suggestions and requests from both reviewers have been addressed adequately, and the manuscript is a nice contribution to the field.

Nice work, compliments!

Reviewer #2: Thank you for the revised text; all comments have been addressed and explained with great care, which is highly appreciated. It has become a fine and relevant article that clearly adds value to practice.

7. PLOS authors have the option to publish the peer review history of their article (what does this mean?). If published, this will include your full peer review and any attached files.

Reviewer #1: No

Reviewer #2: No

---

## [Editor Report · Acceptance letter]

PONE-D-25-16850R1

PLOS ONE

Dear Dr. Cain,

I'm pleased to inform you that your manuscript has been deemed suitable for publication in PLOS ONE. Congratulations! Your manuscript is now being handed over to our production team.

Kind regards,

on behalf of

Dr. Anne E. Martin

Academic Editor

PLOS ONE